# HDAC/σ1R Dual-Ligand as a Targeted Melanoma Therapeutic

**DOI:** 10.3390/ph18020179

**Published:** 2025-01-28

**Authors:** Claudia Giovanna Leotta, Carla Barbaraci, Jole Fiorito, Alessandro Coco, Viviana di Giacomo, Emanuele Amata, Agostino Marrazzo, Giovanni Mario Pitari

**Affiliations:** 1Dream Factory Lab, Vera Salus Ricerca S.r.l., 96100 Siracusa, Italy; 2J4Med Lab, Via Paolo Gaifami 9, 95126 Catania, Italy; 3Department of Drug and Health Sciences, University of Catania, 95125 Catania, Italy; 4Institut Català d’Investigació Química (ICIQ), Avinguda dels Països Catalans 16, 43007 Tarragona, Spain; 5Department of Biological and Chemical Sciences, New York Institute of Technology, Old Westbury, NY 11568, USA; 6Department of Medicine, Columbia University, New York, NY 10032, USA; 7Department of Pharmacy, University “G. d’Annunzio”, Chieti-Pescara, Via dei Vestini 31, 66100 Chieti, Italy

**Keywords:** melanoma, dual-ligands, HDACi, σ_1_ receptor, cancer proliferation, angiogenesis, cell spreading

## Abstract

**Background:** In melanoma, multiligand drug strategies to disrupt cancer-associated epigenetic alterations and angiogenesis are particularly promising. Here, a novel dual-ligand with a single shared pharmacophore capable of simultaneously targeting histone deacetylases (HDACs) and sigma receptors (σRs) was synthesized and subjected to phenotypic in vitro screening. **Methods**: Tumor cell proliferation and spreading were investigated using immortalized human cancer and normal cell lines. Angiogenesis was also evaluated in mouse endothelial cells using a tube formation assay. **Results:** The dual-ligand compound exhibited superior potency in suppressing both uveal and cutaneous melanoma cell viability compared to other cancer cell types or normal cells. Melanoma selectivity reflected inhibition of the HDAC-dependent epigenetic regulation of tumor proliferative kinetics, without involvement of σR signaling. In contrast, the bifunctional compound inhibited the formation of capillary-like structures, formed by endothelial cells, and tumor cell spreading through the specific regulation of σ_1_R signaling, but not HDAC activity. **Conclusions:** Together, the present findings suggest that dual-targeted HDAC/σ_1_R ligands might efficiently and simultaneously disrupt tumor growth, dissemination and angiogenesis in melanoma, a strategy amenable to future clinical applications in precision cancer treatment.

## 1. Introduction

Melanoma is a clinically aggressive cancer arising from the malignant transformation of melanocytes, pigment-producing cells resident at various tissue locations including the skin, mucosae, leptomeninges, eyes and inner ears [1,2]. The main risk factor for malignant melanocyte transformation is exposure to ultraviolet radiation and the progressive accumulation of genetic mutations [3]. The two most common clinical presentations for melanoma are cutaneous melanoma (CM) and uveal melanoma (UM) [4]. Although CM represents less than 10% of all malignant skin tumors, it is the deadliest form of skin cancer, mainly due to its high metastatic potential [4]. As a consequence, patient survival rates at 5 years from diagnosis range from 98% to 16% for localized and metastatic CM, respectively [5,6]. UM accounts instead for only 5% of all primary melanoma cases, and includes melanomas of the choroid, ciliary body and iris of the eye [7]. However, UM is the most common primary intraocular malignancy in adults [7], exhibiting a unique genetic mutational landscape and extremely aggressive behavior [7]. Thus, despite early diagnosis and treatment with conventional therapies, nearly 50% of all UM patients have poor outlooks and develop hepatic metastases within 1 year from diagnosis [8].

Epigenetic reprogramming via histone deacetylation has been identified as a crucial regulator of melanoma progression, driving the tumor mutation burden and the emergence of drug resistance [9]. In this regard, histone deacetylases (HDACs) are a class of enzymes that affect gene expression by removing acetyl groups from histone proteins, a process leading to chromatin condensation and transcriptional repression [10]. HDAC levels are often increased in malignant tumors, including CM and UM, and are inversely associated with favorable clinicopathological parameters and patient survival [11,12]. Moreover, experimental models of melanoma have unequivocally demonstrated HDAC involvement in the pathogenesis of the disease [13,14,15,16]. Presumably, HDAC deregulation in cancer promotes alterations in coordinated transcriptional programs that control key cellular processes, such as cell proliferation, apoptosis, tumor metabolism and immunogenicity [17]. Noteworthy, a variety of HDAC inhibitors (HDACi) are being evaluated in multiple solid and hematologic malignancies, with promising results in CM and UM [18,19,20].

Tumor angiogenesis and invasion are critically permissive factors for metastatic progression in CM and UM [21]. UM cells secrete elevated levels of vascular endothelial growth factor (VEGF), which promotes cancer cell migration, endothelial activation and neovascularization in tumors [21,22]. In addition, emerging targeted antiangiogenic therapies are proving utility in the potential management of patients with CM and UM. Among these, compounds targeting sigma receptors (σRs) are attracting interest for their ability to specifically disrupt motility and angiogenesis-dependent processes in UM cells [23]. A class of unique proteins, σRs are divided into two distinct subtypes: the σ_1_R situated at the mitochondria-associated endoplasmic reticulum (ER) membrane, and the σ_2_R in the ER-resident membrane [24,25] encoded by the *TMEM97* gene [26]. Beyond their involvement in many physiological functions such as memory and brain plasticity, neurotransmission and pain [24], σRs are upregulated in distinct cancer types and may contribute to tumor growth, invasion and angiogenesis [27,28,29].

Previously, dual-ligand prodrugs encompassing HDACi and σ_1_R antagonist/σ_2_R agonist functional profiles were developed [30], which provided potent antiproliferative and antimigratory effects in UM and VEGF-stimulated human retinal endothelial cells [30]. Subsequently, non-esterified multiligand drugs with simultaneous HDAC/σR signaling regulation were obtained and confirmed as a favorable antiproliferative pharmacology in various cancers [23]. In the present work, this latter series of hybrid dual-function ligands was expanded in the quest for a highly potent and selective anti-melanoma drug. Thus, employing an original design, a novel dual-ligand small molecule concurrently targeting HDAC enzymes and σ_1_R was synthesized and subjected to functional characterization with respect to the proliferation and spreading of melanoma cells and endothelial-dependent angiogenic processes.

## 2. Results

### 2.1. Design of the Dual-Target Anti-Melanoma Hybrid

To construct a novel HDAC/σR dual-ligand with meaningful anticancer and antiangiogenic actions, successful strategies of medicinal chemistry were employed, as earlier described [23,30]. Appropriate functional groups were selected to provide an original single pharmacophore capable of simultaneously targeting HDAC enzymes and σRs and anticipated to display excellent binding affinities [23,30]. Specifically, the novel dual hybrid molecule (compound **5c**; original name, MRJF15) was conceived to exhibit three functionally distinct, chemical domains (Figure 1): (a) a σR binding moiety, formed by the hydrophobic region I and the piperidine ring of the σ_1_R antagonist/σ_2_R agonist haloperidol with its conserved nitrogen atom, which plays a crucial role in σR binding; (b) a central hydrophobic linker, comprising the inner benzene of the HDACi LBH-589 (panobinostat); and (c) a zinc binding group (ZBG), represented by the hydroxamic acid group of panobinostat, responsible for chelating the metal Zn^2+^ in the catalytic pocket of HDAC and inhibiting its activity [31,32].

The structural configuration of the new dual-ligand **5c** afforded intentional, synergistic molecular features (Figure 1). Thus, the σR binding moiety, with its hydrophobic portion, might alternatively serve as the capping region (CAP) for the HDACi moiety of the molecule, required for optimal enzymatic inhibition by sealing the access to the catalytic pocket of HDACs. Also, the central benzene could coincidentally fulfill the role of the hydrophobic region II for the σR ligand moiety of the dual-target hybrid [33,34]. Importantly, the selected haloperidol portion should confer specific anti-melanoma and antiangiogenic properties by targeting σR signaling, mimicking its function in dual-target prodrugs [30]. Finally, the HDACi molecular end, formed by hydroxamic acid attached to the central benzene linker, should provide elevated tumor-selective antiproliferative pharmacology, as demonstrated in other HDAC/σR dual-function compounds [23].

In addition, the new dual-target hybrid is predicted to be a general HDACi but a selective σ_1_R ligand [23]. These functional characteristics may shed light on the precise contribution of σR signaling in dual-ligand drug pharmacology. Fundamentally, the new dual-ligand hybrid displayed the critical structures of both the σR-binding haloperidol core and the zinc-chelating nucleus of an established HDACi (Figure 1), rationally assembled in a compacted molecular configuration and providing a single pharmacophore for optimal therapeutic performance [23]. In this way, the design was aimed at obtaining a small molecule (386.88 Da) with improved ligand affinity, target selectivity and biological efficacy.

### 2.2. Chemical Synthesis

The synthetic route for the preparation of the dual-ligand **5c**, an α,β-unsaturated hydroxamic acid derivative with a central benzene linker, involved four reaction steps (Figure 1). In the first step, an esterification reaction was carried out between trans-4-formylcinnamic acid (compound **1**) and methyl iodide in N,N-dimethylformamide (DMF) to obtain (*E*)-3-(4-formylphenyl)methyl acrylate (compound **2**). Subsequently, a reductive amination reaction was performed between compound **2** and 4-(4-chlorophenyl)piperidin-4-ol in tetrahydrofuran (THF), in the presence of sodium triacetoxyborohydride [NaHB(OAc)_3_] and acetic acid, yielding compound **3c**. In the third step, the ester group of compound **3c** was hydrolyzed (in methanol) with 1 M lithium hydroxide (LiOH). Following hydrolysis, the carboxyl group was activated using ethyl chloroformate, in the presence of triethylamine in THF. Then, this activated intermediate underwent a nucleophilic reaction with O-(tetrahydro-2H-pyran-2-yl)hydroxylamine (NH_2_OTHP) in THF, to produce compound **4**. Finally, the deprotection of the hydroxyamino group in compound **4** was achieved by hydrolysis with 0.6 N hydrochloric acid (HCl) aqueous solution, to obtain the final hybrid product **5c**.

### 2.3. Molecular Pharmacology and Tumor Selectivity

Effective target engagements by the novel dual-function agent **5c** were assessed. First, σR binding affinities were determined by employing a radioligand binding assay [23,30]. Dual-targeted **5c** exhibited >100 fold higher affinity for σ_1_R (Ki, 19.0 nM) than σ_2_R (Ki, 2051.2 nM), suggesting the induction of selective σR signaling (Table 1) as for previous congeners [23]. Similarly, compound **3c**, the hydroxamic-null precursor of **5c** (Figure 1) anticipated to lack HDACi activity, also behaved as a selective σ_1_R ligand (Table 1) with superior binding affinities for σ_1_R (Ki, 28.0 nM) than for σ_2_R (Ki, 1330.0 nM). These binding behaviors differentiated **3c** and **5c** from their parental chemical scaffold haloperidol, which bound to both σ_1_R and σ_2_R with high affinities (Table 1) and resembled those of the selective σ_1_R agonist pentazocine [35] (Table 1).

Next, the ability of **5c** to inhibit HDACs was investigated, employing various cancer cell lysates as biological sources of diverse HDAC enzymatic activities [23], including colon (HCT116), breast (MCF7), prostate (PC3) and gastric (AGS) carcinoma cells. First, dose–response curves were generated for the HCT116 cell lysates (Figure 2a), which demonstrated that **5c** reduces HDAC-dependent catalytic activity with a calculated 50% inhibitory concentration (IC_50_) of 14.6 nM. Importantly, HDACi activity by **5c** was profound (~80% inhibition with respect to the vehicle control) and of a similar extent to that of the positive control trichostatin A (Appendix A), a potent and non-selective HDAC inhibitor [36]. The HDAC inhibitory effects of **5c** were generalizable to other cancer cell types, with significant suppression of resident HDAC enzymatic activities compared to respective control conditions, which mirrored those of trichostatin A (Appendix A). Together, the molecular pharmacology studies of receptor engagements indicated that **5c** may signal as a dual-target ligand with selective σ_1_R, but general HDACi actions.

To establish the anti-melanoma selective effects of the novel HDAC/σ_1_R hybrid, complete antiproliferative dose–response curves were generated by employing two in vitro human models, the UM 92-1 cells and normal lung fibroblasts MRC-5 (Figure 2b). The UM 92-1 model was intentionally chosen for these studies because 92-1 cells express all the molecular targets of interest, including HDAC proteins and both σ_1_R and σ_2_R [30,37]. Moreover, they exhibit sensitivity to anticancer actions by dual-target HDAC/σR prodrugs [30]. The lung fibroblasts MRC-5 were in turn interrogated as an experimental surrogate of normal systemic tissues, to ascertain in vitro safety. Of relevance, **5c** appropriately discriminated between tumor and normal cells, with antiproliferative sigmoidal curves shifted to the left in 92-1 cells, compared to MRC-5 cells (Figure 2b). Estimates of antiproliferative IC_50_ confirmed that this hybrid exhibited significantly superior (~5 fold) potency in UM cells, compared to normal lung fibroblasts (Table 2). This latter finding is important because it suggests that **5c** possesses favorable therapeutic profiles, reflecting selective anti-melanoma actions, in the context of safer behaviors in normal cells (Table 2).

### 2.4. Cancer Specificity and Antiproliferative Signaling in Melanoma Cells

The tumor specificity of the test compound was investigated by employing immortalized human cells from four different cancers (CM A375, lung A549, renal 786-O and colorectal T84) and normal skin fibroblast WS1 (Figure 2c). Dual-ligand hybrid **5c** efficiently and selectively reduced proliferation in three additional human cancers (CM, lung and renal adenocarcinomas) compared to normal skin fibroblasts, as indicated by the ~8–10 fold shift to the left in correspondent antiproliferative dose–response curves (Figure 2c). In contrast, colon cancer T84 cells were resistant to **5c** antitumor actions (Figure 2c). IC_50_ calculations confirmed that compound **5c** exhibited higher antiproliferative potencies in melanoma cells (both UM and CM), compared to other normal or cancer cell types (Table 2). Accordingly, a heat map representation of IC_50_ values from all cancer cells demonstrated a melanoma-specific sensitivity to **5c** antiproliferation, with UM 92-1 and CM A375 cells ranking as the first and second most suppressed tumors, respectively, by the multiligand hybrid (Figure 2d).

Importantly, melanoma-specific antiproliferation by **5c** reflected HDAC targeting and suppression of dependent epigenetic regulations, as demonstrated by the loss of activity upon disruption of the HDACi-enabling hydroxamic group (with precursor compound **3c**; Figure 1 and Figure 3). In contrast, exposure of UM (Figure 3a) and A375 (Figure 3b) cells to the σ_1_R agonist pentazocine or the σ_2_R antagonist AC927 [38] did not alter their proliferative kinetics, nor did it perturb **5c**’s antiproliferative actions. Together, these data indicate that HDAC, but not σR, is the main pharmacological target for **5c**-mediated antiproliferation in human melanoma cells.

### 2.5. Effects on Endothelial-Dependent Angiogenic Processes

The ability of the dual-function hybrid **5c** to affect endothelial-driven neoangiogenesis was explored using the tube formation assay with mouse endothelial C166 cells (Figure 4). To focus on the motility and phenotypic organization of capillary-like structures (Figure 4a) [39,40] away from the mechanistic contributions of cell proliferation, short observational periods (3 h) were selected upon C166 cell seeding onto basement membrane-like Matrigel and VEGF-A-stimulation [39]. As expected, VEGF-A activated the endothelial C166 cells and promoted the formation of vasculature-like networks to a greater degree than the vehicle control (Figure 4a,b). Also, the HDAC/σRs bifunctional prodrug (S)-(-)-MRJF22 [(-)-1], used as the positive control [30], significantly inhibited angiogenesis by VEGF-A in these cells (Figure 4a,b). Treatment with **5c** at antiproliferative concentrations (30 nM; Figure 3) was without effects on VEGF-A-stimulated angiogenesis (Figure 4b). In contrast, C166 exposure to **5c** at 5 µM, a concentration level capable of inducing σR-dependent antiangiogenesis [30], abrogated the ability of VEGF-A to induce neovascular networks (Figure 4a), with significant suppression of quantitative angiogenic parameters including tube lengths, mesh areas and node numbers (Figure 4b). Of relevance, the hydroxamic-null compound **3c** mimicked the effects of **5c** on endothelial-dependent angiogenesis (Figure 4b), demonstrating that HDACi is not involved in the disruption of VEGF-induced microvascular organizations by the novel dual-target hybrid.

Application of the σ_1_R agonist pentazocine or the σ_2_R antagonist AC927 inhibited the ability of **5c**, **3c** and (-)-1 to suppress the proangiogenic phenotype of C166 cells, although with different degrees of efficiency with respect to each ligand and the precise angiogenic parameter quantified (Appendix A). A comprehensive examination of **5c**-dependent antiangiogenesis (Figure 5a), achieved by combining all parameters examined in one score, revealed that pentazocine completely prevented **5c**’s effects on vascular tube formations (Figure 5b). AC927 also inhibited **5c**’s actions on angiogenesis, but to a significantly lesser extent than pentazocine (Figure 5b). In contrast, inhibitory activities of VEGF-A-stimulated angiogenesis by (-)-1, a signaling ligand of both σ_1_R and σ_2_R [30], were partially and equally blocked by co-incubations with pentazocine or AC927 (Figure 5). All together, these observations suggest that the HDAC/σ_1_R dual-target **5c** inhibits VEGF-A-stimulated vascular morphogenesis in endothelial C166 cells by acting as a selective σ_1_R antagonist, but without the involvement of HDAC-dependent pathways.

### 2.6. Receptor Signaling in Melanoma Cell Spreading

In solid tumors, changes in cancer cell shape (spreading) underlies migration, invasion and metastasis to distant organs [41]. The impact of the HDAC/σ1R dual-ligand **5c** on locomotory organelle formation (lamellipodia, filopodia) in melanoma cells was investigated with the tumor cell spreading assay [42,43,44]. Quantification of the fraction of UM 92-1 cells extending membrane protrusions revealed that short **5c** exposures (1 h) provoked a shrinkage of this migratory cell population, with a correspondent expansion of cells with rounded morphology (Appendix A). In contrast to antiangiogenic effects (Figure 4), significant inhibition of the invasive cell shape by **5c** was already apparent at low concentrations (100 nM; Figure 6), coincident with those eliciting anti-melanoma proliferation (Figure 3). The HDACi-null analog **3c** also significantly suppressed membrane protrusion-driven 92-1 cell spreading, with nearly identical efficacy and potency as **5c** (Figure 6). Although with similar efficacy, the reference HDAC/σR multiligand prodrug (-)-1, which does not discriminate between σ_1_R and σ_2_R [30], reduced UM cell spreading with seemingly higher potencies (Figure 6).

To establish the role of σ_1_R as the downstream molecular target for the dual-function hybrid **5c**, locomotory-driven cell spreading was then examined in human breast cancer MCF7 cells, which exhibit low levels of endogenous σ_1_R protein and silenced σ_1_R-dependent biological pathways [45]. Treatment with **5c** failed to suppress tumor cell spreading in MCF7 cells (Figure 6). Accordingly, the HDACi-null analog **3c** and the reference prodrug (-)-1 were unable to significantly inhibit MCF7 cell spreading (Figure 6). Together, these findings support the hypothesis that HDAC/σ_1_R dual-target **5c** disrupts locomotory organelle formation in melanoma cells by selectively targeting σ_1_R signaling pathways, without the involvement of HDAC regulation.

## 3. Discussion

Compounds with multi-target pharmacology are emerging as promising small molecules for the safer, more efficacious therapeutic management of multifactorial diseases such as cancer [23,30,31]. In this context, compared to multitherapy with distinct drugs, the simultaneous regulation of different molecular targets, endowed by the dual-ligand strategy, should enable low-dose exposures and improve patient compliance by reducing off-target effects [31]. Indeed, dual-ligands are superior therapeutics anticipated to target distinct biological sites of action in the same tumor cell, thereby maximizing treatment responses. Moreover, this innovative synthetic design is suited to enhance drug therapeutic potentials further, exploiting the synergistic opportunities offered by simultaneously targeting distinct pathways while minimizing the risk of target-based drug resistance [31].

Dual-target drugs could be particularly useful in melanoma, a disease with poor prognosis and high treatment failure rates [1,2,3]. Accordingly, HDAC/σR dual-ligand prodrugs reduced UM cell proliferation and motility while concurrently opposing angiogenic processes stimulated by VEGF [30]. In the present study, a novel dual-function HDAC/σR hybrid was synthesized by employing a pharmacophore fusion approach, wherein the σR ligand part of the molecule was directly attached to the central hydrophobic linker of the selected HDACi moiety (Figure 1). Importantly, the distinct structural components of the parental ligands simultaneously provided reciprocal permissive domains to the dual-target pharmacophore, with the HDACi’s hydrophobic linker acting as the hydrophobic region II for the σR ligand, and the distal hydrophobic end of the σR ligand as the CAP portion for HDACi (Figure 1). In this way, the key functional groups of the precursor drugs contributed to forming a single pharmacophore, simultaneously fitting HDAC and σR whilst maintaining excellent binding interactions.

Importantly, the newly developed dual hybrid **5c** induced potent antiproliferative actions in cancers, which were tumor-selective and specific to UM and CM cells (Table 2). The melanoma selectivity reflected the inhibition of HDAC-dependent epigenetic regulation of proliferative kinetics, without the involvement of σR signaling, as demonstrated by the lack of inhibitory effects using the hydroxamic-null precursor **3c** (Figure 3). Apart from the essential role of ZBG, though, it is conceivable to speculate that additional structural components of the hybrid **5c** strengthen its antiproliferative activity. Among these, the hydrophobic benzene of the central linker (Figure 1) was previously selected as a biologically optimal molecular module, conferring potent anticancer activities on HDAC/σR dual-ligands [23]. Moreover, the CAP region for HDACi, covered in the novel hybrid by its σR moiety (Figure 1), seals the gate of the catalytic pocket and is instrumental in elevated HDAC inhibition.

HDACs comprise an important family of 18 isozymes that can induce profound epigenetic modifications in cells and are being pursued as drug targets for many types of disorders [9]. In cancers, these enzymes regulate the expression and activity of numerous proteins involved in tumor initiation and progression [46]. By removing acetyl groups from histones, HDACs create a non-permissive chromatin conformation that prevents the transcription of specific genes. However, in addition to histones, HDACs can bind to and deacetylate a variety of other protein targets, including transcription factors and other abundant cellular proteins implicated in the control of cell growth, differentiation and apoptosis [9]. As a result, HDACis have recently emerged as promising cancer therapeutics, including in UM and CM [46]. Since the dual-ligand **5c** behaves as a non-selective HDACi in various tumor cells (Appendix A), it is predicated that its role as an anticancer agent could be generalized to other cancers, beyond its specific antiproliferative effects in melanomas.

Of note, the dual hybrid **5c** provided a novel selective σ_1_R ligand, expanding the class of similar dual-ligand congeners [23]. Indeed, **5c** exhibited different σR binding affinities than its chemical scaffold haloperidol, with similar σ_1_R affinity but disrupted σ_2_R binding ability, effects shared by the ester precursor **3c** (Table 1). These original functional outcomes probably reflect the substantial pharmacophore remodeling of the novel dual-ligand compound (and its hydroxamic-null precursor), which comprises a differentially rearranged hydrophobic region II (Figure 1). Moreover, **5c** functioned as a selective σ_1_R antagonist (Figure 5) whose biological activity was antagonized by pentazocine, a putative σ_1_R agonist (Table 1).

The σ_1_R is a unique integral membrane chaperone operating in mitochondria-associated ER domains [25,27]. Acting as a scaffolding protein that allosterically modulates the activity of its associated proteins, σ_1_R is increasingly regarded as a key regulatory signaling receptor implicated in many biological processes [47]. In cancer, σ_1_R ligands inhibited tumor migration, proliferation and survival [47]. Furthermore, σ_1_R has been demonstrated to suppress tumor growth, alleviate cancer-associated pain and exert immunomodulation [30]. Here, the bifunctional **5c** compound inhibited VEGF-stimulated capillary morphogenesis by endothelial cells, through the specific regulation of σ_1_R signaling, but not HDAC (Figure 5). These results are in agreement with the modulation of angiogenesis by σ_1_R in various cell models [46] and support the utility of targeting this receptor to oppose hematogenous tumor cell dissemination.

Selective σ_1_R targeting with the dual-function hybrid **5c** also disrupted invasive locomotory organelle formation in melanoma cells (Figure 6). In this context, invading tumor cells form membrane protrusions (e.g., lamellipodia, filopodia) to spread and migrate into neighboring tissues by extending their leading edges along the direction of the invasive front [43,45]. Thus, the inhibition of σ_1_R functions through pharmacological antagonism could afford the dual-ligand hybrid significant antimetastatic potential in melanoma. Accordingly, the colon cancer cell shape which promotes migration and invasion was suppressed by σ_1_R signaling through the human voltage-dependent K^+^ channel human ether-à-go-go-related gene [48]. Moreover, σ_1_R is often upregulated in tumors [49,50], and its expression potentiated invasive cancer behavior and angiogenesis in vivo, resulting in poor survival [48].

Taken collectively, the present findings suggest that dual-targeted HDAC/σ_1_R ligands could represent a novel paradigm for melanoma therapeutics, being versatile and highly potent. Indeed, they are predicted to efficiently and simultaneously oppose many cancer-permissive pathogenetic programs in melanoma, including tumor proliferation, migration, invasion, angiogenesis and metastasis.

## 4. Materials and Methods

### 4.1. Chemistry

#### 4.1.1. General Procedure

Required chemicals were obtained from Merck (Darmstadt, Germany) or Fluorochem (Derbyshire, UK). All reagents and chemicals were processed, and reactions performed as previously detailed [30]. Flash chromatography purification was done on Merck silica gels 60 (230−400 mesh) and nuclear magnetic resonance spectra (^1^H NMR and ^13^C NMR recorded at 500 MHz) were acquired with Varian INOVA spectrometers, using CDCl_3_, CD_3_OD and DMSO-*d*_6_ with 0.03% of TMS as the internal standard (Appendix A) [30]. High Resolution Mass Spectrometry (HRMS) was conducted using Q-Exactive with ESI source (Thermo Scientific, Waltham, MA, USA). MS full scan acquisitions were performed in positive ion mode, in the m/z range 150.0–800.0. The HCD gas on fragmentation method was used, settled in enhanced resolution, with a maximum injection time of 50 ns (Appendix A).

#### 4.1.2. Synthesis and Product Characterization

Methyl-(2E)-3-(4-formylphenyl)prop-2-enoate (**2**). The synthesis of intermediary **2** was performed as described in previous studies [23]. The obtained solid product (4.6 g) exhibited a yield of 85.4%. ^1^H NMR (500 MHz, CDCl_3_): δ 3.80 (s, 3H), 6.53 (d, J = 16.0 Hz, 1H), 7.63–7.90 (m, 5H), 10.01(s, 1H). ^13^C NMR (125 MHz, CDCl_3_): 53.41, 116.80, 128.52, 129.53, 134.90, 138.15, 142.28, 163.69, 191.22.

Reductive amination. Compound **2** (7.3 mmol) was added (dropwise; RT) to an anhydrous tetrahydrofuran (THF) solution of 4-(4-chlorophenyl)piperidin-4-ol (7.3 mmol) under a nitrogen atmosphere. After addition of AcOH (7.3 mmol), the reaction mixture was stirred (30 min) and added dropwise to a NaBH(OAc)_3_ solution (10.9 mmol) in 1.85 mL of anhydrous THF at RT in a nitrogen atmosphere. The reaction mixture was stirred overnight at RT. Subsequently, a saturated solution of NaHCO_3_ was added, and the aqueous phase was repeatedly extracted with CHCl_3_. The combined organic phases were dried with Na_2_SO_4_ anhydrous, filtered, and evaporated under vacuum. The crude was purified by MPLC using EtOAc as the eluent to obtain intermediate **3** (compound **3c**):

Metil-(E)-3-(4-{[4-(4-clorofenil)-4-idrossipiperidin-1-il]methyl}phenyl)acrylate (**3**, compound **3c**). White solid 4.78 g (yield 73%). ^1^H NMR (500 MHz, CDCl_3_) δ 7.85–7.60 (d, 1H, J = 15.0 Hz), 7.49–7.29 (m, 8H), 6.425 (d, 1H, J = 15.0 Hz), 3.80 (s, 3H), 3,58 (s, 2H), 2.77–2.75 (m, 2H), 2.49–2.44 (m, 2H), 2.15–2.09 (m, 2H), 1.72–1.65 (m, 2H). ^13^C NMR (500 MHz, CDCl_3_) δ 167.51, 146,84, 144.63, 133.24, 132.76, 129.56, 128.38, 128.04, 126.10, 117.35, 71.02, 62.75, 51.69, 49.36, 38.43. HRMS (ESI, m/z) calcd for [C_22_H_25_ClNO_3_]^+^ (M + H)^+^ 386.152, found. 386.144.

Synthesis of the hydroxyamide. An aqueous solution of LiOH 1 M (2 mmol) was added dropwise (at 50 °C) to a MeOH solution of intermediate **3** (compound **3c**; 1 mmol), and the reaction mixture was stirred overnight. After solvent evaporation (in vacuum), residues were dissolved in THF/DMF (1:1) and supplemented first with TEA (2 mmol), and then with an ethyl chloroformate solution (4 mmol) in THF (added dropwise at 0 °C). Following 3 h stirring, O-(tetrahydro-2H-pyran-2-yl)-hydroxylamine (5 mmol) was added under nitrogen at RT, and the reaction stirred overnight. The resulting residue was washed with a saturated solution of NaHCO_3_, and the aqueous phase extracted with CHCl_3_. The combined organic phases were dried with anhydrous Na_2_SO_4_, filtered, and evaporated under vacuum. The crude was purified by MPLC to obtain the intermediate **4**:

(E)-3-(4-{[4-(4-clorofenil)-4-idrossipiperidin-1-il]metil}fenil)-N-(tetrahydro-2H-pyran-2-yl-oxy) acrylamide (**4**). White solid 0.310 g (yield 65%). ^1^H NMR (DMSO-d_6_) δ 11.20 (s, 1H), 7.54–7.34 (m, 9H), 6.49 (d, 1H), 4.91–4.89 (m, 2H), 3.98–3.96 (m, 1H), 3.15 (s, 2H), 2.60–2.56 (m, 2H), 2.45–2.43 (m, 2H), 189–193 (m, 2H), 1.78–1.62 (m, 2H), 1.58–1.54 (m, 6H). ^13^C NMR (DMSO-d_6_) δ 165.55, 162.57, 149.03, 140.71, 139.25, 133.30, 130.75, 129.34, 127.77, 127.43, 126.80, 118.15, 100.98, 69.36, 61.85, 61.35, 48.86, 37.81, 27.77, 24.63, 18.26.

Then, a solution of HCl 1.25 M in EtOH was added (dropwise at 0 °C) to an EtOH solution of intermediate **4**, and the reaction was stirred for 24 h (at RT). After solvent evaporation (in vacuum), the residue was dissolved in EtOH and precipitated with Et_2_O to obtain **5** (compound **5c**):

(E)-4-(4-chlorophenyl)-4-hydroxy-1-{4-[3-(hydroxyamino)-3-oxoprop-1-en-1-yl]benzyl}piperidin-1-ium chloride (**5**, compound **5c**). White solid 72 mg (yield 88%). ^1^H NMR (DMSO-*d*_6_) δ 11.24 (broad s, 1H), 7.73–7.41 (m, 9H), 6.55 (d, 1H, J = 15 Hz), 4.37–4.36 (m, 2H), 3.26–3.20 (m, 4H), 2.50 (s, 2H), 1.75–1.78 (m, 2H). ^13^C NMR (DMSO-*d*_6_) δ 162.36, 146.95, 137.34, 135.84, 132.02, 131.242, 130.86, 128.00, 127.62, 126.55, 120.32, 67.94, 58.22, 47.68, 34.65. Anal. calcd. for C_21_H_23_ClN_2_O_3_⸱HCl: C, 65.20; H, 7.24; N, 5.99; Found: C, 65.34; H, 6.98; N, 6.17. HRMS (ESI, m/z) calcd for [C_21_H_24_ClN_2_O_3_]^+^ (M + H)^+^ 387.148, found. 387.139.

### 4.2. Cell Cultures and Reagents

Unless otherwise specified, cell lines were obtained from the American Type Culture Collection (ATCC; Manassas, VA, USA). Human UM cell line 92-1 was maintained at 37 °C (5% CO_2_) in RPMI-1640 medium, containing 10% fetal bovine serum (FBS), 2 mM L-glutamine, 100 units/mL penicillin and 100 µg/mL streptomycin. Human renal adenocarcinoma cell line 786-O (ATCC^®^ CRL-1932™) was cultured in RPMI with 10% FBS. Human cutaneous melanoma cells A375 (ATCC^®^ CRL-1619™), human lung adenocarcinoma A549 (ATCC^®^ CCL-185™) and mouse endothelial cell C166 (ATCC^®^ CRL-2581™) were cultured in Dulbecco’s modified Eagle’s medium (DMEM) with 10% FBS, whereas human skin fibroblast WS1 (ATCC^®^ CRL-1502™) and human lung fibroblast MRC-5 (ATCC^®^ CCL-171™) in EMEM with 10% FBS. Human colorectal carcinoma cell line T84 (ATCC^®^ CCL-248™) was cultured in DMEM/F12 medium with 10% FBS. Human breast cancer MCF7 (HTB-22™) and human colorectal carcinoma cell line HCT116 (CCL-247™) were maintained in DMEM-high glucose (Euroclone, Milan, Italy) supplemented with 10% FBS and 1% penicillin/streptomycin. Human gastric adenocarcinoma AGS cells and human prostate adenocarcinoma PC3 cells were purchased from the European Collection of Authenticated Cell Cultures (ECACC, Porton Down, Salisbury, UK) and maintained in HAM’s F12 medium supplemented with 10% FBS and 1% of penicillin/streptomycin. Cell media and general reagents were from Euroclone S.p.A. (Pero, Milan, Italy). The positive control (-)-1 was prepared as previously described [30]. Radioactive (+)-Pentazocine (Italian Minister of Health permit to produce and use SP/072 5 April 2019) and AC927 were prepared according to published methods [51,52].

### 4.3. Cell Proliferation

Cells were seeded in 96-well plates at a cell-specific density, following quantification of cell numbers on counting chambers after trypsinization and staining with trypan blue. After seeding (72 h), at a cell confluence of ~50%, the medium was removed, and treatments were started with fresh media. Treatments, carried out in quadruplicate, lasted for 48 h. Then, the cells were fixed with 4% paraformaldehyde in PBS for 20 min and, after washing, stained with either 1% crystal violet in water or an acridine orange solution (50 µg/mL) for an additional 20 min. At the end of the incubations, cells were washed (3 times) with 100 μL/well of bidistilled water. Finally, for crystal violet assays, 100 μL of 10% acetic acid was added to each well, and the absorbance (λ = 590 nm) was measured with a Synergy HTX Bioteck spectrophotometer (BioSPX, Beersel, Belgium). For acridine orange assays, cell staining was evaluated by measuring the fluorescence (excitation 485/20 nm, emission 528/20 nm) with the Synergy HTX Bioteck reader.

### 4.4. Radioligand Binding Assay

The *σ*_1_R and *σ*_2_R binding studies were performed as previously described [30,53,54], using guinea pig brain membranes and appropriate radioligands, including [^3^H]-(+)-pentazocine (45 Ci/mmol) for *σ*_1_R, and [^3^H]-1,3-di-o-tolylguanidine ([^3^H]-DTG; 31 Ci/mmol) for *σ*_2_R. Nonspecific binding was also assessed with unlabeled haloperidol or DTG. Radioactivity was determined with a Beckman LS6500 scintillation counter. Inhibition constants (Ki values) were calculated with the EBDA/LIGAND program (Elsevier/Biosoft).

### 4.5. HDAC Activity Assay

HDAC activity was determined in the indicated cell lysates (50 µg proteins; 37 °C) using the non-selective HDAC Activity Colorimetric Assay kit (BioVision, Cambridge, UK), following the procedures described in previous studies [23]. Samples were read at 405 nm with a microplate reader (Multiskan GO (Thermo Fisher Scientific, Waltham, MA, USA)), and the HDAC activity was expressed as the relative optical density (O.D.) values. The IC_50_ value of compound **5c** (0–20 µM) in HCT116 cells was calculated using the GraphPad Prism^®^ software (version 10.4.1).

### 4.6. Tube Formation Assay

The tube formation assay using mouse endothelial C166 cells was analyzed employing Matrigel (BD, Bedford, MA, USA), as previously described [30]. Briefly, 96-well plates coated with Matrigel (50 μL/well) were filled with 100 μL medium/well and maintained overnight. Endothelial cells were seeded at 2.5 × 10^4^ cells/well and incubated (for 3 h) with VEGF-A (80 ng/mL), in the presence of the indicated treatments. Then, capillary-like structures were imaged (×4 magnification) with a CCD camera connected with an inverted EVOS microscope. Finally, ImageJ software (NIH, Bethesda, MD, USA) was employed to quantify tube formations.

### 4.7. Cell Spreading

Adherent 92-1 and MCF-7 cells were harvested using 0.05% Trypsin-EDTA and plated in 6-well plates. After 1 h incubations with the indicated treatments, the number of cells that formed, or not, distinct protrusions (lamellipodia, filopodia) were scored with an inverted microscope (20×, EVOS cell imaging system, Thermo Fisher, Waltham, MA, USA). Results were reported as either the fraction of total cell number per microscopic field (≥400 cells evaluated for each condition) or the relative percentages of inhibition of cell spreading with respect to the vehicle controls.

### 4.8. Statistical Analysis

Unless otherwise specified, data were expressed as means ± SEM of ≥3 independent experiments, each conducted at least in duplicate. Student’s *t*-test or one-way ANOVA multiple comparisons test were employed for all statistical analyses. Differences between groups were considered significant at *p* values < 0.05.

## 5. Conclusions

A novel dual-target HDAC/σR hybrid was synthesized and investigated herein, exhibiting a single pharmacophore which simultaneously regulated HDAC and σ_1_R signaling in cells. The dual-ligand preferentially suppressed melanoma cell proliferation and membrane protrusions by inhibiting HDAC-mediated epigenetic regulations and σ_1_R activation, respectively. Occurrence of these effects in the same malignant cells, afforded by the dual-target strategy, is anticipated to provide synergistic opportunities for melanoma-specific eradication therapies. The novel bifunctional agent also restricted the formation of capillary-like structures in endothelial cells, an effect mediated by the selective regulation of σ_1_R, and not HDAC. Future studies should explore the mechanisms of action for the dual-ligand, focusing on molecular events downstream to target engagements that underlie individual biological outcomes in tumors.

The ability to concurrently disrupt many malignant processes, including tumor growth, invasion and angiogenesis, clearly represents an advantageous trait for the dual-function HDAC/σ_1_R ligand. In this way, it might provide the elusive curative, safe remedy that would shrink tumor volumes at primary sites, prevent cancer dissemination to distant organs and defeat the occurrence of tumor resistance. Translation of this therapeutic paradigm into clinical applications would contribute to the advancement of precision medicine in oncology.

## Data Availability

All data supporting the findings of the study are available from the corresponding authors, G.M.P. and A.M., upon reasonable request.

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
