# Peer review of "HDAC/σ1R Dual-Ligand as a Targeted Melanoma Therapeutic"

_pharmaceuticals, 2025, doi:10.3390/ph18020179_

Round 1

Reviewer 1 Report

Comments and Suggestions for Authors

Claudia Giovanna Leotta et al submitted a research work titled, “HDAC/σ1R dual-ligand as targeted melanoma therapeutic”. Authors aimed to establish dual ligand targeting simultaneously histone deacetylases (HDACs) and sigma receptors. Authors synthesized and characterized the ligands and evaluated for their anticancer potential by dual targeting strategy. This is quite interesting and paving a new pathway towards viewing of the role of dual targeting and inhibition on multiple pathways in cancer cell cycle as this research is emerging recently (as per reference 23). The introduction part of the manuscript covered all the aspects related to the work. All the figures are presented in good resolution and the results were discussed, accordingly. The conclusion is drawn as per the outcome of the results and proposed for future experiments. The manuscript can be revised majorly before the acceptance as mentioned below:-

1.     In Figure 1, author can write “LBH-589” as ‘Panobinostat’ (this is an established non-selective HDAC inhibitor).

2.     Is Halperidol tested earlier against “HDAC inhibition’. If so, mention in the introduction section.

3.     The nomenclature of compounds mentioned as 3c and 5c, why not to mention ‘3’ and ‘5’?

4.     In Scheme 1: (iii) 1. LiOH 1M, pls justify.

5.     HRMS data missing for all the synthesized compounds needs to be included. If not for atleast for compounds 3c and 5c.

6.     Authors used standard drugs in Table 1 (Haloperidol and Pentazocine (selective σ1R agonist): both are CNS active and established as antipsychotic and analgesic agents. Why authors not selected established HDAC inhibitors as standard anticancer drugs for the comparison of the synthesized compounds representing the category of HDAC and σ1R agents.

7.     In Materials and methods: If synthesis and characterization data already reported in literature, then no need to repeat as plagiarism occurred. Hence, authors can mention that the compounds so and so prepared and characterized as per published literature and mention the reference number 23.

8.     Sections 4.4, 4.5 showed plagiarism as they may be an established protocol. I would suggest mentioning concisely with citing respective references.

9.     The conclusion and future directions are missing that needed be mentioned before the section 4.  

Reviewer 2 Report

Comments and Suggestions for Authors

This study significantly contributes to precision cancer therapeutics through an innovative dual-ligand design targeting HDACs and σ1Rs for melanoma treatment. Here are a few minor comments to be addressed.

1.     Clarification on selectivity to be included: As the isoform selectivity assays for HDAC were not performed, how could readers access the selectivity of the developed HDAC/σ1R dual ligand?

2.     Page 7, line 273: The study indicates that the antiproliferative concentration, 30 nM, is not sufficient for the inhibition of angiogenesis but effective at 5 µM. Can authors elaborate on the term “σR-dependent antiangiogenic concentration” for better clarity?

3.     As HDAC inhibition assay using 5c was also carried out on MCF7, PC3, AGS, and gastric adenocarcinoma cells, it can be incorporated in the same sentence on page 5, line 180.

For clarity

4.     Page 2, line 79: “were develop” could be modified to “were developed” or the sentence can be re-written to ensure clarity.

5.     Page 4, line 156: The authors mentioned the term “Radioligand binding assay,” whereas the same assay in the methodology section (Page 14, line 554) was mentioned as “Receptor binding studies”. To ensure clarity for the readers, could authors follow any of the terms in both places?

6.     Include the full name of the reagent DTG (page 14, line 565).

7.     Cite a reference for pentazocine as an σ1R agonist (page 4, line 164).

8.     Cite a reference for the chosen HCT116 cells for HDAC inhibition assay (page 5, line 178).

9.     Cite a reference for AC927 as an σ2R antagonist (page 6, line 242).

10.  Check the figure citation on page 9, line 312. “Figure S3” – Figure S3 describes the NMR spectra and not the “cells with rounded morphology”.

11.  Can authors cite the “Cell Spreading counting table S1” at an appropriate position in the in-text of the main manuscript to enhance the clarity of the results?

12.  Can authors include the abbreviations as a heading with all the terms being used in the manuscript?

Reviewer 3 Report

Comments and Suggestions for Authors

A novel dual-ligand compound targeting histone deacetylases (HDACs) and sigma receptors (σRs) was synthesized and demonstrated to selectively suppress melanoma cell proliferation, spreading, and angiogenesis in vitro, with distinct mechanisms of action for each target. This dual-targeting strategy underscores the potential of HDAC/σ1R ligands in disrupting melanoma progression and angiogenesis, presenting a relevant approach for precision cancer therapy. The study acknowledges several limitations, which are thoroughly discussed by the authors throughout the article.

Multiple assays were conducted to evaluate the dual mechanism of action; however, in several instances, duality was not conclusively confirmed. Nevertheless, the presented findings remain significant for readers of Pharmaceuticals, and the work could be considered for publication following minor revisions.

Line 92: "Successful strategies were employed, underlying the construction of HDAC/σR dual-ligand compounds with meaningful anticancer and antiangiogenic efficacies": Sentence is unclear.

Line 156: "σR binding affinities were calculated employing" - affinities were determined by radioligand binding assay (and calculation was only the final step of the whole process)

Line 157: "Ki, 18.97 nM" falsely accurate measurement, it should be rounded to "Ki, 19 nM" - the same goes for all similar data (e.g. 24.85 ± 5.83 should be reported as 25 ± 6)

Line 442: "(1H NMR and 13C NMR recorded at 200 or 500 MHz) were obtained on Varian INOVA spectrometers" - are these data correct?

Round 2

Reviewer 1 Report

Comments and Suggestions for Authors

The authors have addressed all the comments and the manuscript can be accepted in its present form.